# Exploring the sentiment of entrepreneurs on Twitter

**James Waters[1], Nicos Nicolaou[1], Dimosthenis Stefanidis[2], Hariton Efstathiades[2¤], George Pallis [2]\*, Marios Dikaiakos[2]**

**1** Warwick Business School, University of Warwick, Coventry, United Kingdom, **2** Department of Computer Science, University of Cyprus, Nicosia, Cyprus

¤ Current address: PricewaterhouseCoopers (PwC), Nicosia, Cyprus
\* gpallis@cs.ucy.ac.cy

## Abstract

Sentiment analysis is an evolving field of study that employs artificial intelligence techniques to identify the emotions and opinions expressed in a given text. Applying sentiment analysis to study the billions of messages that circulate in popular online social media platforms has raised numerous opportunities for exploring the emotional expressions of their users. In this paper we combine sentiment analysis with natural language processing and topic analysis techniques and conduct two different studies to examine whether engagement in entrepreneurship is associated with more positive emotions expressed on Twitter. In study 1, we investigate three samples with 6.717.308, 13.253.244, and 62.067.509 tweets respectively. We find that entrepreneurs express more positive emotions than non-entrepreneurs for most topics. We also find that social entrepreneurs express more positive emotions, and that serial entrepreneurs express less positive emotions than other entrepreneurs. In study 2, we use 21.491.962 tweets to explore 37.225 job-status changes by individuals who entered or quit entrepreneurship. We find that a job change to entrepreneurship is associated with a shift in the expression of emotions to more positive ones.

**Data Availability Statement:** The paper has 2 studies; the data and the code for Study 1 can be found at https://figshare.com/s/41203e4f316cdffa609d. Study 2 involves third-party data that the authors cannot legally distribute.

## Introduction

Many studies have found that emotions–by which we mean positive or negative feelings experienced when considering a specific topic, person, or object [1–3]–influence behaviours and activities that are central to entrepreneurship. For example, emotions have been shown to influence intentions to work towards a goal [4–6], leadership behaviours [7], trust [8], communication [9], personal influence [9], performance [10], negotiation outcomes [11, 12], information collection about uncertain environments [13, 14], and perceptions of change [15].

There is also increasing evidence on the emotions experienced by entrepreneurs. For example, Baron et al. [16] propose that there is a curvilinear relationship between entrepreneurs' positive affect and the tasks related to the development of the venture, while Tata et al. [17] find that entrepreneurs show positive affect more often than non-entrepreneurs do, and negative affect less often. Patzelt and Shepherd [18] also find that self-employed individuals experience negative affect less often than the general population. Jennings et al. [19] examine the

Researchers can request access to the data at crunchbase.com. The authors had no special access to the data and researchers can request access to the data in the same way the authors obtained it.

**Funding:** This work was partially supported by the iSocial EU Marie Curie ITN project (FP7-PEOPLE-2012-ITN). No additional funding was received for this work.

**Competing interests:** No competing interests exist.

effects of emotional arousal in generating novel solutions, while Spivack, McKelvie and Haynie [20] show how the emotional aspects of the entrepreneurial experience can lead to a "behavioural addiction to entrepreneurship". A meta-analysis on the emotional side of entrepreneurship finds a positive relationship between positive affect and entrepreneurial performance [21]. Further evidence on the importance of emotions in entrepreneurship is provided in the insightful studies of Cardon et al. [22, 23], which argue that emotion towards entrepreneurial identity motivates an entrepreneur's choice of career and activities.

Building on these works, we argue that engagement in entrepreneurship brings a sense of autonomy to entrepreneurs and places them in a position to create a stimulating work environment, raising their sense of well-being [24, 25]. This sense can be manifested in the expression of positive emotions even if the objects of those emotions are unrelated to entrepreneurship [26, 27]. In this work, we examine differences in the emotional expression of entrepreneurs versus non-entrepreneurs as manifested in their messages ("tweets") on Twitter, the major online platform for public expression. We apply emotion mining, a subtask of sentiment analysis [28–30], to determine the polarity of emotions in tweets. Sentiment and emotion are closely related concepts of Natural Language Processing with sentiment reflecting the emotion that 'colors' an expressed opinion or idea. Sentiment analysis techniques deduce a writer's emotions and opinions through Natural Language Processing. Applied on Twitter, sentiment analysis algorithms take the posted text, emoticons and hashtags of each tweet and return a score ranging from -1 (extreme negative emotions) to 1 (extreme positive emotions). Text is the main medium of Internet-mediated human communication and, therefore, analysing the sentiment and the topic of tweets at a massive scale can provide important observations regarding the emotions that groups of users express on Twitter, either in general or about specific topics. A taxonomy of sentiment analysis and a complete survey on emotion theories is presented in Yadollahi et al. (2017) [31].

In this paper we present two different studies: the first study investigates three different samples of 6,717,308, 13,253,244 and 62,067,509 tweets sent by entrepreneurs and non-entrepreneurs from two major centres of innovation in Europe and U.S. (London and Los Angeles) and globally respectively. The second study examines 21,491,962 tweets sent by individuals who engaged in or quit entrepreneurship, as documented by 37,225 job-status changes identified in Crunchbase, a leading online business-related database.

Our work complements earlier studies using Twitter data in managerial or entrepreneurial studies [17, 32–34] and makes a number of contributions:

Our first contribution is to examine whether there are differences between categories of entrepreneurs (traditional, social and serial entrepreneurs) in their emotional expression on social media platforms like Twitter. Most of the previous research focused on entrepreneurs' emotions but not on the emotional expression of entrepreneurs in social media platforms. It is valuable to understand this expression because it can influence recipients–both by communicating information about the entrepreneur's views [13] and by aligning the recipient's and the entrepreneur's emotional states [14]. In addition, our study expands previous work that considers how emotion influences social entrepreneurship intention and participation [35–41] and shows that social entrepreneurship can lead the entrepreneur to express more positive emotions than other forms of entrepreneurship on social media platforms. We also contribute to the limited literature on emotions in serial entrepreneurship [42, 43] by showing that serial entrepreneurship can lead to the expression of less positive emotions than other forms of entrepreneurship. Finally, the study of social media content with big data techniques allows us to conduct analyses at a much larger scale.

Our second contribution is to compare differences in emotions expressed on Twitter between entrepreneurs and non-entrepreneurs in relation to different topics. There is

currently limited work examining the effect of entrepreneurship on emotions relating to topics unconnected with entrepreneurship. It is worth examining this link because emotions can affect an entrepreneur's cognition and judgements [2], which can lead them to act rashly in areas in which they have no expertise. We build on prior literature that has considered emotions specific to certain topics, such as Cardon et al.'s [22] and Gielnik et al.'s [44] examination of emotions towards entrepreneurial identity and activity. By employing 10 different topics, namely "Society", "Recreation", "Health", "Business", "Home", "Science", "Computers", "Games", "Arts", "Sports, we show that entrepreneurs express significantly more positive emotions than non-entrepreneurs in all topics.

Our third contribution is to examine how transition from entrepreneurship to non-entrepreneurship and vice versa can affect the emotions of an individual on social media platforms responding to recent calls [45, 46]; we show that engagement in entrepreneurship is associated with a higher emotional positivity. These findings further support the argument that entrepreneurship is associated with more positive emotional expressions on social media platforms, and demonstrate that, controlling for the characteristics of the entrepreneur, entrepreneurial actions are associated with increased positivity. Finally, our study contributes to recent research that has examined the digital footprints of entrepreneurs using computerized text analyses [47–49].

## Theory and research questions

### Emotions of entrepreneurs and non-entrepreneurs

This subsection compares the emotions of entrepreneurs and non-entrepreneurs on social media platforms. Our arguments are based on the idea that entrepreneurship gives the entrepreneur freedom to choose their work environment and objectives. This freedom brings the entrepreneur a sense of autonomy which is psychologically beneficial [50]. Additionally, the choices they make can create a working environment that is more stimulating, and both of these factors can raise the entrepreneur's emotion. Our argument also relates to affective events theory, which has examined work settings as proximal causes of emotional reactions [51, 52]. Affective events theory has been described as the seminal contribution on the role of affect in organizations [53] and has informed not only our understanding of affect in organizations but also other theoretical perspectives, including psychological contracts, organizational justice, and work stress [54].

To begin, we argue that the act of choosing her work aims and conditions can be enjoyable in itself for an entrepreneur. Entrepreneurs make a much wider range of choices than employees. Whereas an employee may be offered a precise set of tasks, workplace conditions and working hours as an employment package, an entrepreneur chooses them individually with far greater freedom. The entrepreneur's act of selecting their own employment conditions with little external pressure can bring them a sense of autonomy [50, 55], which can be psychologically beneficial, reducing anxiety, raising self-esteem, and bringing enhanced well-being [56, 57].

Further, the choices made by entrepreneurs may also make their emotions more positive. Entrepreneurs are well-informed about their own preferences, and motivated to act to achieve them. As a result, their chosen work environment and duties are more likely to be stimulating, at least relative to employed work where employees have less input into its design. Entrepreneurs are more likely to feel greater motivation and well-being while undertaking their duties [58]. When entrepreneurs work in activities they consider valuable, they can feel intense positive emotions from the engagement [22].

An entrepreneur's positive emotions are likely to be shown in their social media. Additionally, an entrepreneur's more positive emotions can appear in their evaluations of people [2], events [59], and potential outcomes of actions [60]. If they then describe their evaluations in their posts, the positivity will again be shown. Of course, an entrepreneur may hide their positive emotions on social media; however, they will usually express or even magnify them, for the following reason. If an easy and costless act motivates their workers and potential investors to support their activities, an entrepreneur will usually do it. When an entrepreneur expresses positive emotions, they often influence other people to have positive emotions [61], making the people more likely to evaluate the entrepreneur and their activities well [2, 60], and so provide support [61]. Thus, an entrepreneur will usually express their positive emotions on social media.

In summary, the first Research Questions (RQs) are the following ones:

*RQ1*: *Are entrepreneurs more likely than non-entrepreneurs to exhibit positive emotions on social media platforms*?

*RQ2*: *How does a job change from entrepreneur to non-entrepreneur and vice versa affect the emotions of an individual on social media platforms*?

## Emotions of social entrepreneurs

Our next research question looks at social entrepreneurs, people who apply business practices to address social problems or create social value (there are numerous variations on this definition [62–65]). As both traditional entrepreneurs and social entrepreneurs have substantial freedom to set their own goals and working conditions, the arguments that we presented on why traditional entrepreneurs show more positive emotions continue to apply to social entrepreneurs. Beyond those arguments, we can also note that social entrepreneurs may receive gratitude from others for their activities and gain social esteem. However, it isn't clear beforehand that these factors would affect social entrepreneurs more than traditional entrepreneurs. Accordingly, we frame our research question without rigid prior theorizing, and later revisit our theory in the light of the observed evidence.

We investigate the following Research Question (RQ):

*RQ3*: *Are social entrepreneurs more likely than other entrepreneurs to exhibit positive emotions on social media platforms*?

## Emotions of serial entrepreneurs

This subsection considers the emotions on social media platforms of serial entrepreneurs—entrepreneurs who establish multiple businesses over time [66, 67]. Research has found that serial entrepreneurs are likely to be characterised by obsessive passion, with the start-up assuming a disproportionate space in an entrepreneur's identity [42]. As Thorgren and Wincent [42] argue, "rather than the individual having control over their engagement in the activity, the activity controls the individual." In addition, Podoynitsyna et al. [43] found that emotional reactions by entrepreneurs on strategic issues changed significantly as they became serial entrepreneurs.

We argue that serial entrepreneurs may be less likely to show positive emotions than other entrepreneurs. We start by proposing that a serial entrepreneur's experience leads to memories of entrepreneurial failures that lower the positivity of their emotions. Serial entrepreneurs are "hardened" through time by the difficulties of starting multiple ventures. We then argue that

serial entrepreneurs experience fewer pleasures associated with novel experiences than new entrepreneurs, lowering their emotions in general.

In their careers, many serial entrepreneurs would have set up businesses that failed: a substantial percentage of newly established companies cease trading within a few years of establishment, whether through bankruptcy or another form of discontinuance [68–70], which can trigger grief, anger, or frustration [71, 72]. Experiences from those times would often have been unpleasant, and memories of the events can reinforce this feeling through mood-congruent selective encoding, whereby only events associated with unpleasant emotions are retained in memory [2, 73]. A serial entrepreneur may reflect on these unpleasant prior experiences [2], which are less likely to be considered by less experienced entrepreneurs and which impact negatively the serial entrepreneur's emotions.

Additionally, many of the tasks undertaken by serial entrepreneurs are repeated with each new business they found. Such repeated tasks include regulatory permission to set up a company, undertaking a hiring process for all company positions, and seeking finance. They may become routine or even monotonous, and serial entrepreneurs may then derive limited pleasure from them. In contrast, new entrepreneurs may see them as a change of routine, a thrill, a source of adventure, a way of alleviating boredom, and a surprise [74], and so may experience greater pleasure from them [75]. As a result, a serial entrepreneur is less likely than other entrepreneurs to show positive emotions.

In summary, we investigate the following Research Question (RQ):

*RQ4*: *Are serial entrepreneurs less likely than other entrepreneurs to exhibit positive emotions on social media platforms*?

## Methodology

To address the above research questions we conducted two studies where we explore the sentiment of entrepreneurs on Twitter platform, since Twitter has prevailed as the major online platform of public expression. The first study addresses the research questions RQ1, RQ3, RQ4, whereas, the second study focuses on RQ2.

## Study 1

In study 1, our sample data sets should take into account only tweets that have been sent by individual users, both entrepreneurs and non-entrepreneurs, from their personal accounts. This means that we have to remove from the dataset tweets sent by Twitter accounts that do not correspond to individual users, and which could bias our analysis. These accounts are either bots (software that autonomously performs actions such as tweeting, retweeting, liking, following, unfollowing, or direct messaging other accounts) or are linked with company or professional profiles, which are mainly used to advertise their owner and are clearly differentiated from the accounts of individuals [76, 77]. To remove these accounts, we evaluated several different profile features (including the number of Twitter friends and followers, the number and frequency of tweets, and reciprocal relationships) which have been studied in the literature and act as the key factors for distinguishing individual Twitter users from other users [78, 79].

Additionally, we used the Botometer tool [76, 80–82], which uses machine learning to extract more than a thousand features (i.e., users and friends meta-data, tweet content and sentiment, network patterns, and activity time series) and its goal is to classify a Twitter account as bot or human. Experimental results have shown that Botometer achieves 95% accuracy [82]. We then used these characteristics and the Botometer tool as the basis for excluding non-individual users.

**Table 1. Number of users, initial tweets, and usable tweets.**

| Raw London Data | | Raw LA Data | | Raw WW Data | |
|---|---|---|---|---|---|
| from January 2013 to January 2015 | | from January 2013 to January 2015 | | from January 2013 to January 2015 | |
| Users: 132,272 | | Users: 350,637 | | Users: 36,907,407 | |
| Tweets: 232,331,077 | | Tweets: 532,738,302 | | Tweets: 12,762,535,499 | |
| **Find entrepreneurs** (based on keywords) | *Entre*: 1,786 | **Find entrepreneurs** (based on keywords) | *Entre*: 3,919 | **Find entrepreneurs** (based on keywords) | *Entre*: 24,573 |
| | *Entre Tweets*: 3,579,691 | | *Entre Tweets*: 7,261,749 | | *Entre Tweets*: 39,641,296 |
| **Select randomly 1,786 non—entrepreneurs** | *Non-Entre*: 1,786 | **Select randomly 3,919 non—entrepreneurs** | *Non-Entre*: 3,919 | **Select randomly 24,573 non—entrepreneurs** | *Non-Entre*: 24,573 |
| | *Non-Entre Tweets*: 3,137,617 | | *Non-Entre Tweets*: 5,991,495 | | *Non-Entre Tweets*: 22,426,213 |

We collected information on tweets and Twitter users—respecting the platform's terms of use and users' privacy—from three different samples: London (132,272 Twitter users and 232,331,077 tweets), Los Angeles (350,637 Twitter users and 532,738,302 tweets) and a world-wide sample (36,907,407 Twitter users and 12,762,535,499 tweets) (see Table 1). We examined London and Los Angeles as they are among the most successful regions in the world in attracting start-ups and venture investors with high rates of entrepreneurial growth and success in attracting funding. Both geographic areas are leading centres of innovation and entrepreneurial hubs (https://www.inc.com/minda-zetlin/best-cities-for-entrepreneurs-austin-dallas-la-san-diego-miami.html; https://www.crowdspring.com/blog/startups-entrepreneurs-best-startup-cities-us/#losangeles; https://londonlovesbusiness.com/london-is-still-top-for-entrepreneurship/#:~:text=London%20continues%20to%20be%20the,the%20UK's%20high%2Dgrowth%20companies.&text=However%2C%20growth%20levels%20among%20young%20enterprises%20are%20down; https://www.wired.com/insights/2014/12/london-innovation) [83] in Europe and the United States, respectively. Furthermore, we selected the specific regions as users from these geographic areas are fluent in English and thus the misspellings in the tweets could be less. We identified 24,573 entrepreneurs in our worldwide sample, 1,786 entrepreneurs in our London sample and 3,919 entrepreneurs in our Los Angeles sample. Finally, we selected three random samples of non-entrepreneurs from the geographic regions under investigation, of equal size to the respective samples of entrepreneurs, to be used for analysis and comparison.

## Variables

**Emotion.** We capture emotion using the *average sentiment of each user* based on all of their published tweets. The emotion in the tweets was measured using VADER (Valence Aware Dictionary for sEntiment Reasoning) [84], a software specifically attuned to sentiment analysis in microblog-like contexts (i.e., Twitter). VADER takes as input the posted text, emoticons and hashtags of tweets and returns a score indicating the negativity or positivity of a tweet. This score ranges from -1 (extreme negativity) to 1 (extreme positivity).

The sentiment score of a tweet in VADER is obtained by summing up the intensity of each word in the text. The mapping of each word to its sentiment score is constructed by examining well-established sentiment word-banks [85–89]. These are combined with general rules that embody grammatical and syntactical conventions for expressing and emphasizing sentiment intensity. In addition, numerous lexical features common to sentiment expression in micro-blogs are incorporated, including a full list of emoticons (for example, ":-)" denotes a "smiley face" and generally indicates positive sentiment), sentiment-related acronyms (e.g., LOL), and commonly used slang with sentiment value (e.g., "nah", "meh" and "giggly"). In a recent

review of emotion and attitude classification software, VADER was shown to achieve the best overall classification accuracy and was able to outperform 22 commonly used tools, including even proprietary/paid sentiment analysis tools like LIWC [90].

Before utilizing VADER tool for finding the sentiment of each tweet, we followed several preprocessing steps. Specifically, we removed URLs, numbers (i.e. numerical text), common symbols (e.g. "=", except the main punctuations e.g.?!;.,'"), jargon symbols or text (e.g. "&"). Afterwards, we replaced 3 (or more) duplicate characters (e.g. "tooool" = > "tool"). Then, we replaced the contractions with the original words (e.g. "I'm" -> "I am"). Also, VADER can handle typical use cases of negation like "not good", "wasn't very good" etc., and it is not affected by cases where negation flips the sentiment of the text.

While our study does not focus on any specific positive or negative emotion (e.g. anger, happiness, etc.), it is important to note that the emotion literature argues that different positive emotions (e.g., cheerfulness) or different negative emotions (e.g., anger, sadness) may have different orthogonal structures [91–93], different informative values [94, 95], and different cognitive determinants [96].

In this study, we focus on two emotional states of a tweet: positive and negative. Specifically, as a dependent variable we use the VADER's score that indicates the negativity or positivity of a tweet. VADER incorporates word-order sensitive relationships between terms and is able to determine the magnitude of intensity through punctuation, capitalization, degree modifiers, negations, slang etc. The output of VADER are the positive, negative, and neutral ratios of sentiment. VADER's score ranges from -1 (extreme negativity) to 1 (extreme positivity).

VADER score: $-1 \leq$ Negative sentiment (e.g. -0.3) $< 0$ (Neutral Sentiment) $<$ Positive sentiment (e.g. 0.2) $\leq 1$

Furthermore, in cases where a tweet contains positive and negative emotions at the same time, VADER sums up the strength/intensity of the sentiment for each word in the text and finds the more dominant emotion. For example, if a sentence contains equal intensities scores of positive and negative emotions, then the output of VADER is 0. Overall, VADER's score determines the emotion intensity in a continuous scale based on the emotional changes in a sentence, and not just the binary polarity (i.e., either positive or negative).

While our analysis assumes that publicly expressed emotions broadly reflect an entrepreneur's underlying emotions, following Tata et al. [17] and Wolfe and Shepherd [89], entrepreneurs may also use communications to manage the impression that others have of them, which can be a key influence in whether they are funded [97].

**Tweet topic.** In order to identify the topic of each tweet in our datasets, we submit every tweet to the uClassify Topics web service, via its online HTTP REST Application Programming Interface. uClassify [98] is an open source machine learning web service that contains several pre-trained text classifiers. This classifier categorizes an unstructured text into 10 general topics, namely: arts, business, computers, games, health, home, recreation, science, society and sports. It is well suited for both short and long texts (tweets, Facebook statuses, blog posts, product reviews etc). From a technical point of view, uClassify comprises a Naïve Bayes classifier that has been trained on 2.8 million documents with data from Twitter, Amazon product reviews and movie reviews. These topics (categories) are from the Open Directory Project [99]. For any given text, uClassify Topics API returns the probability (range of values from 0 to 1) that a specific tweet belongs to a specific topic. Thus, the topic with the highest probability is most likely the topic that the specific tweet should belong to. In order to increase the precision of the uClassify tool and the confidence of the results (assigned topics), we keep only the tweets with topic probability higher than 0.9.

**Entrepreneur.** Entrepreneurs are identified as users who have in their personal Twitter description any of the following terms: "entrepreneur", "founder", "co-founder", "business-

owner", "business owner", "start-up", or "start up". The remaining user profiles from Twitter were classified as non-entrepreneurs. The variable was coded as "1" for entrepreneurs and "0" for non-entrepreneurs.

**Social entrepreneur.** Social entrepreneurs are identified as users who describe themselves as such in their Twitter profile, using the keywords "social" and "entrepreneur". The variable was coded as "1" for social entrepreneurs and "0" for other entrepreneurs.

**Serial entrepreneur.** Serial entrepreneurs are identified as users who describe themselves as such in their Twitter profile, using the keywords "serial" and "entrepreneur". The variable was coded as "1" for serial entrepreneurs and "0" for other entrepreneurs.

**Followers.** The variable "Followers" indicates the number of other users who follow the user. In Twitter, someone can choose to follow a user, meaning that they receive all of the user's messages. This is a measure of the popularity one has in the Twitter network and could be positively associated with emotions [100].

**Followings.** The variable "Followings" indicates the number of other users that the user follows. It is a measure of a user's interest and integration in Twitter and could be positively associated with emotions [100].

**Retweets.** This variable indicates the proportion of tweets per user that are retweets. A tweet is considered as a retweet when a user resends a message that was originally sent on Twitter by another user. This is a measure that quantifies the information diffusion in Twitter and could be associated with emotions [101].

**Geotagged tweets.** This variable indicates the proportion of tweets per user that are geo-tagged. A geotagged tweet carries metadata with the specific latitude and longitude of the location from which it was sent. It has been observed that there are statistically significant demographic differences between users who do or do not geotag their tweets [102].

**Total number of tweets.** This is the number of tweets and retweets a user has published. It depicts a user's engagement on Twitter.

**Hashtags.** The variable "Hashtags" indicates the total number of hashtags divided by the number of tweets per user. Specifically, a hashtag is a short word or phrase which other users can search for through the Twitter interface in order to easily find messages carrying the hashtag. Users make use of hashtags to increase the visibility of their tweets.

**Android source.** This variable indicates the proportion of tweets per user that are generated and published using the Twitter mobile application for Android smartphones.

## Results

Tables 2–4 report the correlations between the variables for the three samples (at the user level of analysis).

Fig 1 plots the overall percentage of positive, negative and neutral tweets both for entrepreneurs and non-entrepreneurs for the three different geographical regions examined (at the tweet level of analysis). As we can observe, the Twitter streams of entrepreneurs contain tweets that are substantially more positive than the tweets of non-entrepreneurs across all samples.

Fig 2 plots the percentage of positive tweets published per day of the week, for entrepreneurs and non-entrepreneurs (tweet level). As we can observe, for both user categories, the percentage of positive tweets is lower during weekends, while it increases during weekdays, reaching highest values towards the end of the week. In all cases, entrepreneurs are consistently more positive than non-entrepreneurs for each day of the week.

Next, we analyse the emotional score of the two groups during extended periods of time (tweet level). Fig 3 plots the emotional score of each calendar day during a period of 2 years;

**Table 2. Descriptive statistics and correlations for London data (N = 3,572).**

| Variables | Mean | S.D. | (1) | (2) | (3) | (4) | (5) | (6) | (7) | (8) |
|---|---|---|---|---|---|---|---|---|---|---|
| (1) Sentiment | .15 | .08 | | | | | | | | |
| (2) Entrepreneur | .50 | .50 | .24*** | | | | | | | |
| (3) Num of followers | 2393.57 | 21687.85 | .04** | .08*** | | | | | | |
| (4) Num of followings | 902.59 | 2367.65 | .10*** | .09*** | .14*** | | | | | |
| (5) Total number of tweets | 6715.92 | 12187.35 | -.09*** | .01 | .07*** | .25*** | | | | |
| (6) Android source | .09 | .21 | -.12*** | -.11*** | -.02 | -.02 | .07*** | | | |
| (7) Retweets | .22 | .16 | .08*** | -.00 | .02 | .04** | .06*** | .08*** | | |
| (8) Geotagged tweets | .14 | .15 | -.12*** | -.16*** | -.04** | -.06*** | -.01 | .02 | -.13*** | |
| (9) Hashtags | .43 | .40 | .24*** | .16*** | .01 | .06*** | -.12*** | -.06*** | .08*** | -.02 |

Notes. *p < .05

** p < .01

*** p < .001.

**Table 3. Descriptive statistics and correlations for the Los Angeles data (N = 7,838).**

| Variables | Mean | S.D. | (1) | (2) | (3) | (4) | (5) | (6) | (7) | (8) |
|---|---|---|---|---|---|---|---|---|---|---|
| (1) Sentiment | .16 | .11 | | | | | | | | |
| (2) Entrepreneur | .50 | .50 | .31*** | | | | | | | |
| (3) Num of followers | 7650.40 | 46232.89 | .08*** | .09*** | | | | | | |
| (4) Num of followings | 3787.92 | 18959.58 | .08*** | .10*** | .66*** | | | | | |
| (5) Total number of tweets | 7873.19 | 20286.14 | -.06*** | .07*** | .24*** | .28*** | | | | |
| (6) Android source | .11 | .25 | -.11*** | -.12*** | -.02* | -.03** | -.01 | | | |
| (7) Retweets | .22 | .19 | -.13*** | -.16*** | -.02* | -.04*** | .01 | .12*** | | |
| (8) Geotagged tweets | .06 | .13 | -.03** | .01 | -.04*** | -.05*** | .02 | .00 | -.10*** | |
| (9) Hashtags | .40 | .48 | .24*** | .26*** | .01 | .03** | .01 | -.08*** | -.04*** | .14*** |

Notes.

*p < .05

** p < .01

*** p < .001.

**Table 4. Descriptive statistics and correlations for worldwide data (N = 49,146).**

| Variables | Mean | S.D. | (1) | (2) | (3) | (4) | (5) | (6) | (7) | (8) |
|---|---|---|---|---|---|---|---|---|---|---|
| (1) Sentiment | .13 | .09 | | | | | | | | |
| (2) Entrepreneur | .50 | .50 | .14*** | | | | | | | |
| (3) Num of followers | 3209.81 | 75810 | .01 | .02*** | | | | | | |
| (4) Num of followings | 860.45 | 4376.67 | .05*** | .08*** | .13*** | | | | | |
| (5) Total number of tweets | 5090.22 | 13931.5 | -.03*** | .03*** | .07*** | .22*** | | | | |
| (6) Android source | .04 | .12 | -.02*** | -.01** | -.00 | -.01 | .08*** | | | |
| (7) Retweets | .13 | .16 | .05*** | .12*** | .01 | .02*** | .13*** | .23*** | | |
| (8) Geotagged tweets | .02 | .06 | -.01** | .06*** | -.00 | .01 | .05*** | .03*** | .01** | |
| (9) Hashtags | .28 | .34 | .09*** | .13*** | .00 | .07*** | .08*** | .09*** | .29*** | .10*** |

Notes.

*p < .05

** p < .01

*** p < .001.

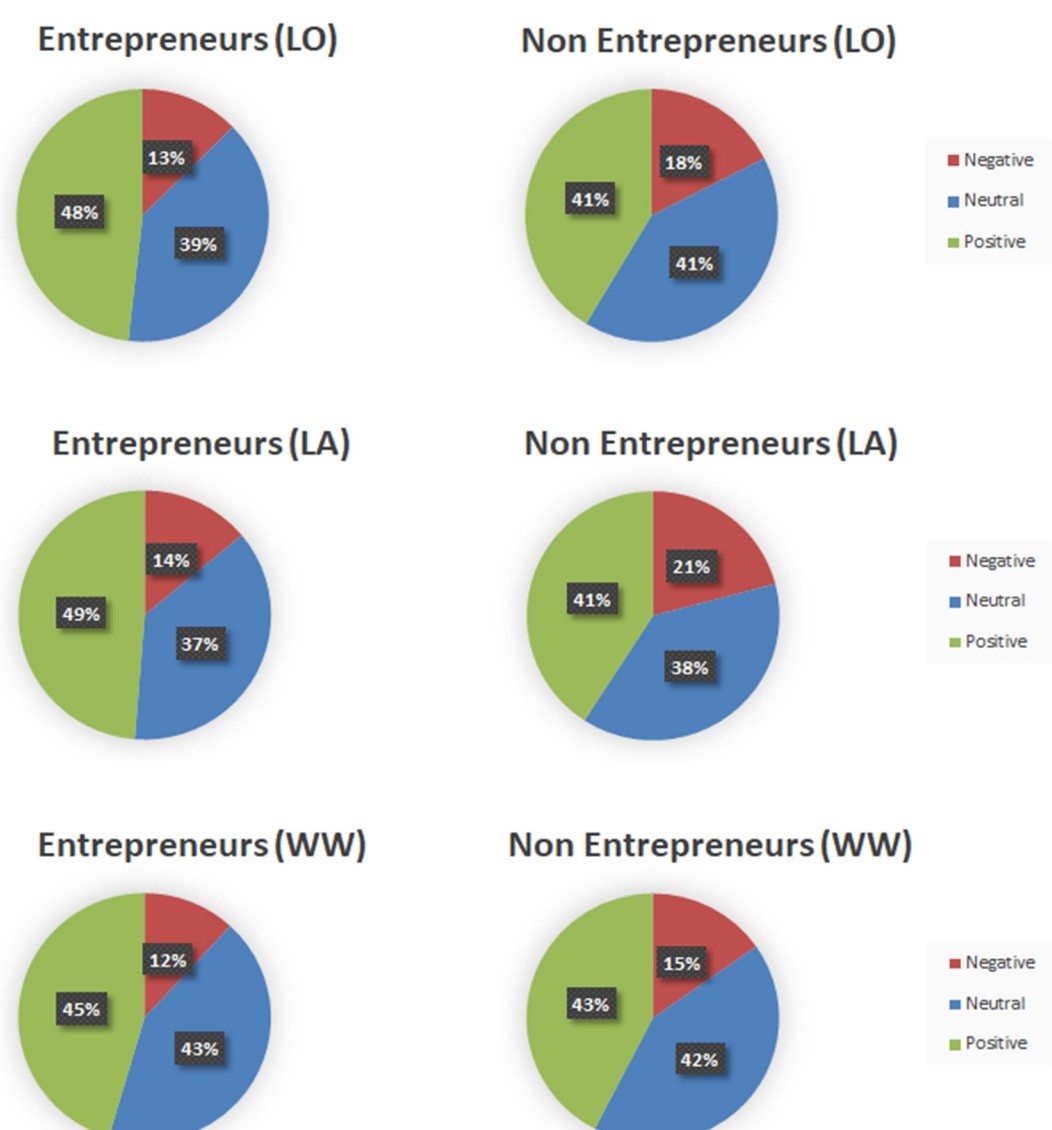

**Fig 1. Tweet emotional comparison between entrepreneurs and non-entrepreneurs for London (top), Los Angeles (middle) and worldwide (tweet level).**

entrepreneurs are generally more positive than non-entrepreneurs throughout the two-year period.

The regression results are shown in Table 5 (at the user level of analysis). The dependent variable is the average sentiment of a user's tweets. Columns 1, 2, and 3 show the results for the London, Los Angeles, and worldwide samples, respectively. The coefficient of "Entrepreneur" is positive and significant in all samples (p<0.001), indicating that entrepreneurs are more likely to exhibit positive emotions than non-entrepreneurs in social media platforms, answering the first research question (RQ1).

We next examine the emotions of entrepreneurs and non-entrepreneurs in different topics. For the classification of tweets into different categories we use uClassify Topics API. Due to service restrictions of uClassify we submitted for classification the tweets of a randomly chosen

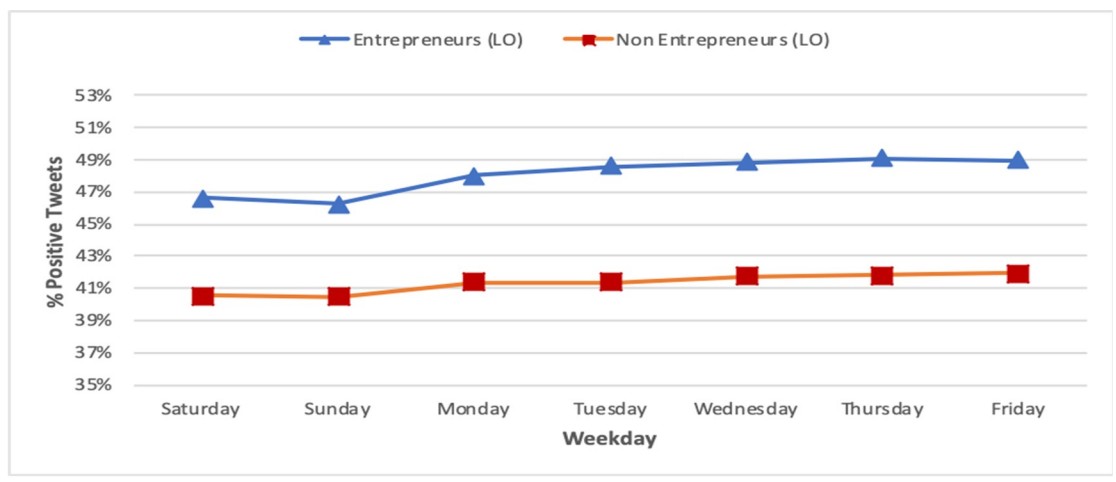

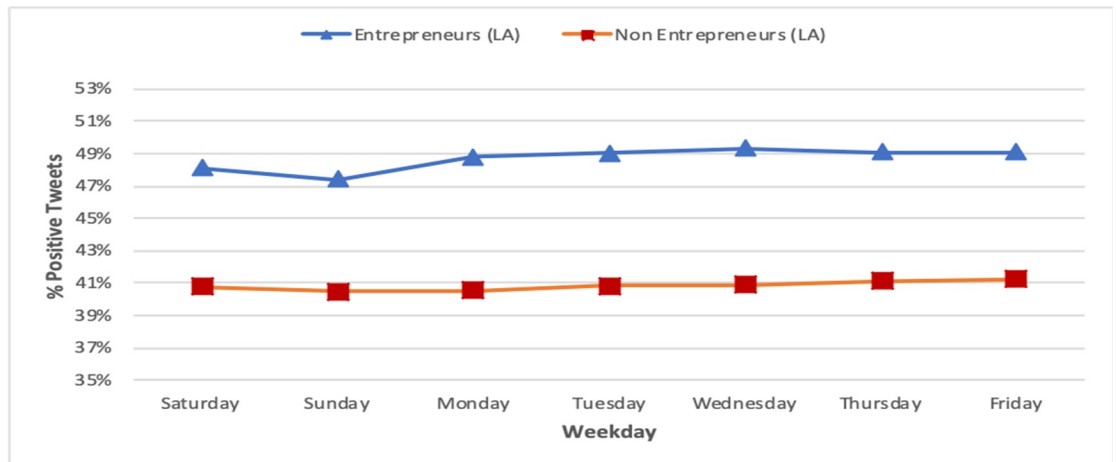

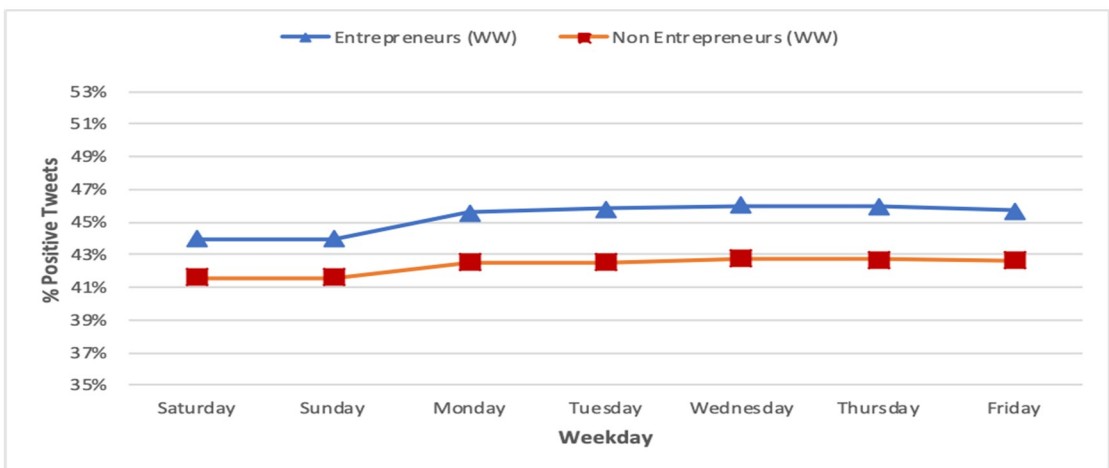

**Fig 2. Tweet emotional comparison per weekday between entrepreneurs and non-entrepreneurs for London (top), Los Angeles (middle) and worldwide (tweet level).**

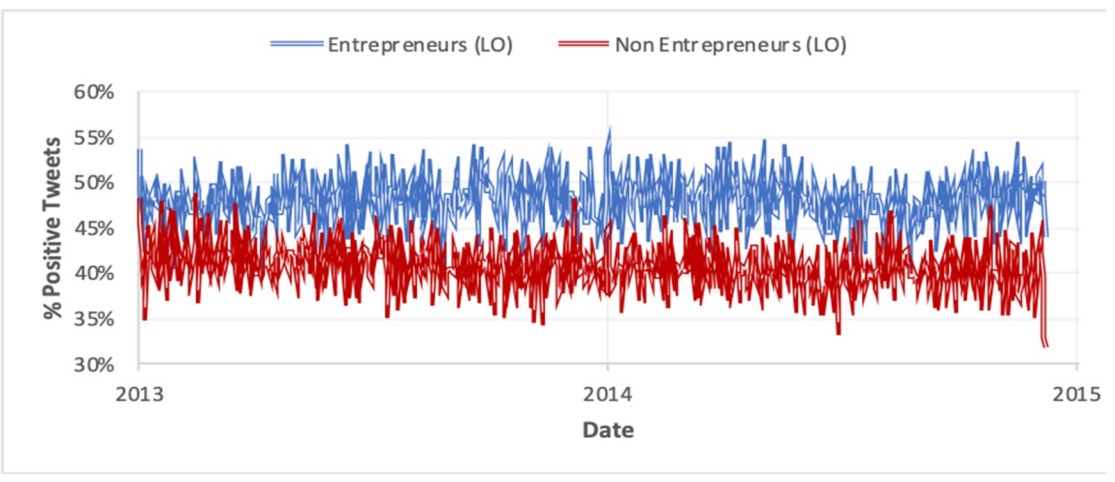

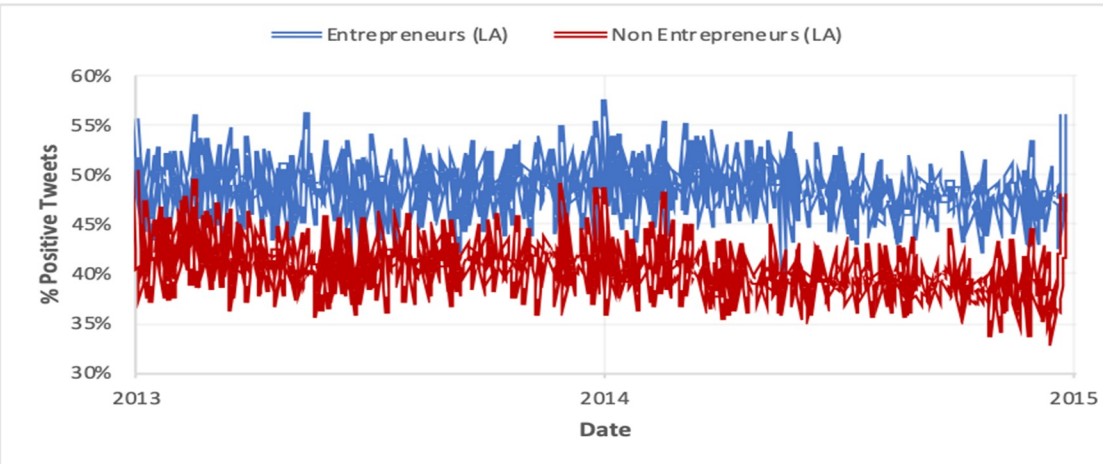

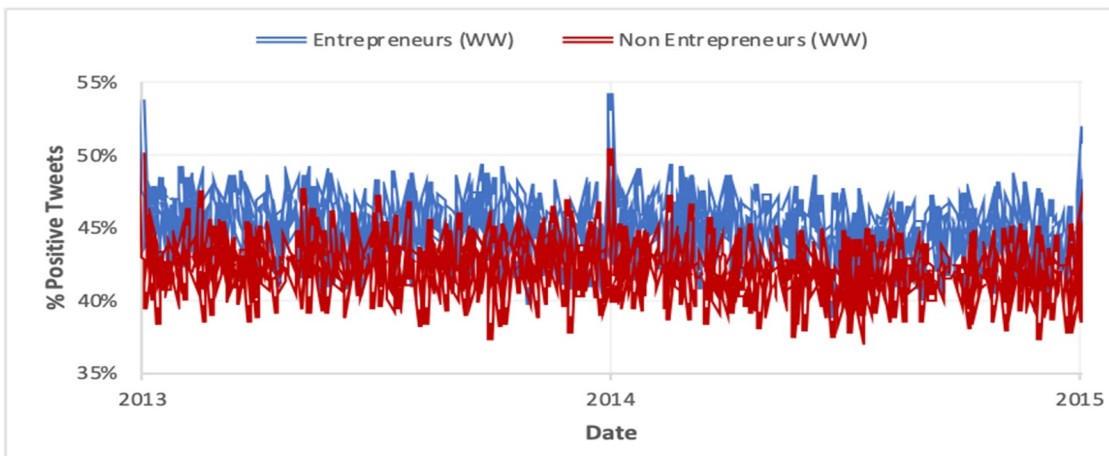

**Fig 3. Tweet emotional comparison per calendar day between entrepreneurs and non-entrepreneurs for London (top), Los Angeles (middle) and worldwide (tweet level).**

**Table 5. Regressions.**

| Sample | All | All | All | All | Entrepreneurs only |
|---|---|---|---|---|---|
| Region | London (1) | Los Angeles (2) | London and Los Angeles (3) | Worldwide (4) | Worldwide (5) |
| Entrepreneur | .029087*** | .054790*** | .047109*** | .023026*** | |
| | (.002572) | (.002507) | (.001899) | (.000844) | |
| Social Entrepreneur | | | | | .007030*** |
| | | | | | (.001680) |
| Serial Entrepreneur | | | | | -.005534 |
| | | | | | (.002998) |
| Number of followers | .000041 | .000128*** | .000113*** | 4.000e-06 | 3.00e-06 |
| (000s) | (.000058) | (.000034) | (.000029) | (6.000e-06) | (8.09e-06) |
| Number of followings | .002887*** | .000186* | .000269*** | .001012*** | .000719*** |
| (000s) | (.000549) | (.000085) | (.000075) | (.000098) | (.000081) |
| Total number of tweets | -.000620*** | -.000541*** | -.000523*** | -.000353*** | -.000274*** |
| (000s) | (.000106) | (.000061) | (.000052) | (.000031) | (.000035) |
| Android source | -.030269*** | -.024918*** | -.025788*** | -.022225*** | -.025443*** |
| | (.005877) | (.004822) | (.003846) | (.003520) | (.003805) |
| Retweets | .033477*** | -.048475*** | -.031764*** | .014975*** | .002395 |
| | (.007980) | (.006284) | (.005052) | (.002735) | (.002831) |
| Geotagged tweets | -.037853*** | -.052155*** | -.052945*** | -.039125*** | -.064773*** |
| | (.008317) | (.009549) | (.006564) | (.006547) | (.006462) |
| Hashtags | .036426*** | .041800*** | .041932*** | .020425*** | .010796*** |
| | (.003158) | (.002577) | (.002047) | (.001290) | (.001291) |
| Constant | .124126*** | .138578*** | .135881*** | .116284*** | .144714*** |
| | (.003198) | (.002571) | (.002050) | (.000706) | (.000693) |
| N | 3,572 | 7,838 | 11,410 | 49,146 | 24,546 |
| R-squared | .131 | .147 | .137 | .030 | .0148 |
| F | 67.31*** | 169.7*** | 226.83*** | 192.1*** | 41.00*** |

Notes. The dependent variable is average sentiment based on the Twitter user's tweets. Standard errors are in parentheses.

*p < .05

** p < .01

*** p < .001.

Column 5 (Social and Serial Entrepreneurs): We excluded 27 users who were both social and serial entrepreneurs.

3-month period (September-November 2014) with 348,349 and 246,793 tweets posted by entrepreneurs and non-entrepreneurs respectively (Table 6).

Tables 7–9 present the sentiment comparison per topic between entrepreneurs and non-entrepreneurs for the 3 datasets (London, Los Angeles and worldwide). In addition, several independent t-tests were run for the 3 datasets to determine if there were differences in the emotion of entrepreneurs and non-entrepreneurs based on each topic. The results showed that for all datasets, entrepreneurs are significantly more positive than non-entrepreneurs in most topics.

Regarding the analysis for social and serial entrepreneurs, we used only the entrepreneurs from the worldwide sample (24,573 entrepreneurs) (We did not use the other London and Los Angeles samples for the analysis of social and serial entrepreneurs as these sample sizes were relatively small to identify large numbers of social and serial entrepreneurs). We identified the social (1,601 profiles) and serial entrepreneurs (473 profiles) (see Table 10) through their Twitter profile and compared them with the rest of the entrepreneurs from the worldwide dataset.

**Table 6. Number of initial and usable tweets for extracting the discussion topic.**

| London Data | | LA Data | | WW Data | |
|---|---|---|---|---|---|
| from September 2014 to November 2014 | | from September 2014 to November 2014 | | from September 2014 to November 2014 | |
| Initial Tweets: 1,005,408 | | Initial Tweets: 2,420,966 | | Initial Tweets: 3,184,087 | |
| Remove tweets that contain only urls, user mentions or symbols | *Tweets*: 951,528 | Remove tweets that contain only urls, user mentions or symbols | *Tweets*: 2,252,311 | Remove tweets that contain only urls, user mentions or symbols | *Tweets*: 3,071,587 |
| Remove tweets with low topic probability score (the highest class probability must be bigger than 0.9) | *Tweets*: 79,822 | Remove tweets with low topic probability score (the highest class probability must be bigger than 0.9) | *Tweets*: 210,997 | Remove tweets with low topic probability score (the highest class probability must be bigger than 0.9) | *Tweets*: 304,323 |

Note: We also compared the sentiment of entrepreneurs versus non-entrepreneurs per topic using all tweets. Our results remained robust and qualitatively similar. However, we noticed that tweets with topic probability $\geq 0.9$ have a lower sentiment than tweets with topic probability $< 0.9$. To investigate this, we compared the grammatical structure of the tweets. Specifically, we first identified the grammatical terms in each tweet (e.g. number of verbs, nouns, etc.) by using NLTK POS tagger (Python library). Then, we compared these grammatical terms before and after filtering the tweets that have topic probability $\geq 0.9$. The results show that tweets with topic probability $\geq 0.9$ have more nouns and fewer verbs, adjectives, and adverbs than tweets with topic probability $< 0.9$. This is expected for topic classification as more nouns improve the topics' semantic coherence [103]. On the other hand, sentiment analysis tools (e.g. VADER) focus more on verbs, adjectives and adverbs [84, 104, 105] as nouns are usually treated as semantically empty (i.e. neutral polarity) [106, 107].

Table 5 column 5 reports our results for social entrepreneurs (research question 2—RQ2) and serial entrepreneurs (research question 3—RQ3). The coefficient of social entrepreneurship is positive and significant (p<0.001), indicating that social entrepreneurs experience a higher emotional positivity than other entrepreneurs. On the other hand, the coefficient of serial entrepreneurship is negative but not significant. This means that we cannot argue that "serial entrepreneurs are less likely than other entrepreneurs to exhibit positive emotions".

Also, we compared social and serial entrepreneurs. (We excluded 27 users who were both social and serial entrepreneurs.) The results show that social entrepreneurs are significantly more likely to exhibit positive emotions than serial entrepreneurs (p<0.001).

## Robustness tests

In order to strengthen our analysis, we performed a series of robustness tests. As we observed earlier, entrepreneurs tend to exhibit more positive emotions than non-entrepreneurs for the majority of topics. However, one could argue that entrepreneurs and non-entrepreneurs may be tweeting about different issues or sub-topics under the same topic, and that this difference may affect the observed difference in emotions between entrepreneurs and non-entrepreneurs.

**Table 7. Two-sample t-test to compare the mean emotional score per topic between entrepreneurs and non-entrepreneurs (London data).**

| Topics | Tweets of Non-Entrepreneurs | | | Tweets of Entrepreneurs | | | t | df | Sig. (2-tailed) |
|---|---|---|---|---|---|---|---|---|---|
| | N | Mean | SD | N | Mean | SD | | | |
| Society | 6539 | 0.044 | 0.742 | 6647 | 0.123 | 0.739 | -6.111 | 13184 | 0.000 |
| Recreation | 2648 | 0.259 | 0.639 | 2430 | 0.310 | 0.667 | -2.773 | 5076 | 0.006 |
| Health | 2089 | 0.090 | 0.751 | 2951 | 0.166 | 0.767 | -3.461 | 5038 | 0.001 |
| Business | 519 | 0.237 | 0.615 | 1636 | 0.328 | 0.639 | -2.861 | 2153 | 0.004 |
| Home | 1993 | 0.241 | 0.678 | 2358 | 0.372 | 0.688 | -6.284 | 4349 | 0.000 |
| Science | 622 | 0.164 | 0.634 | 1060 | 0.251 | 0.625 | -2.742 | 1680 | 0.006 |
| Computers | 2084 | 0.263 | 0.686 | 5995 | 0.279 | 0.693 | -0.889 | 8077 | 0.374 |
| Games | 515 | 0.169 | 0.698 | 566 | 0.223 | 0.689 | -1.272 | 1079 | 0.204 |
| Arts | 5546 | 0.252 | 0.719 | 6578 | 0.355 | 0.683 | -8.057 | 12122 | 0.000 |
| Sports | 16005 | 0.255 | 0.708 | 11041 | 0.287 | 0.694 | -3.716 | 27044 | 0.000 |

Table 8. Two-sample t-test to compare the mean emotional score per topic between entrepreneurs and non-entrepreneurs (Los Angeles data).

| Topics | Tweets of Non-Entrepreneurs | | | Tweets of Entrepreneurs | | | t | df | Sig. (2-tailed) |
|---|---|---|---|---|---|---|---|---|---|
| | N | Mean | SD | N | Mean | SD | | | |
| Society | 15910 | 0.053 | 0.758 | 21261 | 0.117 | 0.777 | -7.984 | 37169 | 0.000 |
| Recreation | 5299 | 0.215 | 0.653 | 7081 | 0.293 | 0.626 | -6.673 | 12378 | 0.000 |
| Health | 4589 | -0.061 | 0.776 | 6994 | 0.117 | 0.799 | -11.870 | 11581 | 0.000 |
| Business | 1193 | 0.194 | 0.604 | 4726 | 0.294 | 0.614 | -5.036 | 5917 | 0.000 |
| Home | 5323 | 0.203 | 0.588 | 7668 | 0.347 | 0.613 | -13.429 | 12989 | 0.000 |
| Science | 1825 | 0.128 | 0.601 | 2649 | 0.188 | 0.643 | -3.167 | 4472 | 0.002 |
| Computers | 4709 | 0.318 | 0.692 | 10897 | 0.275 | 0.664 | 3.651 | 15604 | 0.000 |
| Games | 1626 | 0.156 | 0.717 | 1625 | 0.225 | 0.684 | -2.783 | 3249 | 0.005 |
| Arts | 17359 | 0.211 | 0.741 | 16050 | 0.303 | 0.685 | -11.768 | 33407 | 0.000 |
| Sports | 38777 | 0.213 | 0.731 | 35436 | 0.241 | 0.709 | -5.221 | 74211 | 0.000 |

Table 9. Two-sample t-test to compare the mean emotional score per topic between entrepreneurs and non-entrepreneurs (worldwide data).

| Topics | Tweets of Non-Entrepreneurs | | | Tweets of Entrepreneurs | | | t | df | Sig. (2-tailed) |
|---|---|---|---|---|---|---|---|---|---|
| | N | Mean | SD | N | Mean | SD | | | |
| Society | 20943 | 0.093 | 0.741 | 28766 | 0.163 | 0.695 | -10.821 | 49707 | 0.000 |
| Recreation | 5968 | 0.249 | 0.633 | 9226 | 0.303 | 0.624 | -5.201 | 15192 | 0.000 |
| Health | 5991 | 0.131 | 0.747 | 10584 | 0.211 | 0.699 | -6.860 | 16573 | 0.000 |
| Business | 2372 | 0.247 | 0.617 | 10136 | 0.304 | 0.612 | -4.077 | 12506 | 0.000 |
| Home | 5970 | 0.277 | 0.667 | 7595 | 0.295 | 0.662 | -1.605 | 13563 | 0.109 |
| Science | 3483 | 0.194 | 0.620 | 6637 | 0.244 | 0.599 | -3.895 | 10118 | 0.000 |
| Computers | 21727 | 0.261 | 0.668 | 57823 | 0.292 | 0.653 | -5.902 | 79548 | 0.000 |
| Games | 2372 | 0.175 | 0.717 | 2864 | 0.287 | 0.675 | -5.834 | 5234 | 0.000 |
| Arts | 14882 | 0.310 | 0.700 | 19032 | 0.337 | 0.666 | -3.574 | 33912 | 0.000 |
| Sports | 27915 | 0.227 | 0.715 | 40037 | 0.239 | 0.701 | -2.204 | 67950 | 0.028 |

Table 10. Number of social entrepreneurs and serial entrepreneurs.

| Entrepreneurs—WW Dataset | | Entrepreneurs—WW Dataset | |
|---|---|---|---|
| Entre: 24,573 | | Entre: 24,573 | |
| Entre Tweets: 39,641,296 | | Entre Tweets: 39,641,296 | |
| **Find social entrepreneurs** (based on keyword "social" and "entrepreneur") | *Profiles*: 1,601 | **Find serial entrepreneurs** (based on keyword "serial" and "entrepreneur") | *Profiles*: 473 |
| | *Tweets*: 3,032,714 | | *Tweets*: 768,712 |

To explore this, we randomly chose two topics for which we conducted an in-depth terminology analysis. In particular, we selected all the tweets from entrepreneurs and non-entrepreneurs categorized in the topics of "Religion" and "Health", tokenized them, removed stop words (namely, commonly used terms with no significant semantic weight, such as "the", "and", "or") and compared the remaining content utilizing two different methods from information retrieval. The first method that we use relies on monograms comparison: we take the terms that have been used in the tweets of entrepreneurs and compare them with the corresponding terms used by non-entrepreneurs. The results show that the two groups use similar

terms when talking about "Religion" and "Health", with the similarity in the top-100 most frequent terms reaching 81% and 73% for the London dataset, and 72% and 57% for the Los Angeles dataset, respectively.

Then, we proceed in n-grams analysis where we compare the similarity regarding the position of the characters in a sentence, and not only the appearance of a term. We take again the tweets that lie in the topics of "Religion" and "Health", and proceed in applying n-grams comparison. Again, the results suggest that entrepreneurs and non-entrepreneurs use similar terminology: for the topics of "Religion" and "Health" the n-gram similarity reaches 69% and 61% for the London dataset, and 62% and 52% for the Los Angeles dataset, respectively. These similarity metrics provide a good indication that there is little difference in what entrepreneurs and non-entrepreneurs discuss over Twitter that could affect the emotional score of the two groups' tweets.

We also re-run the regressions using a broader description of serial and social entrepreneurs. Specifically, we re-identified social entrepreneurs as users who have in their personal Twitter description the keyword "entrepreneur" and any of the following terms: "social", "social good", "social venture", "non-profit", "philanthropist" or "philanthropy" (N = 1732). We re-identified serial entrepreneurs as users who have in their personal Twitter description the keyword "entrepreneur" and any of the following terms: "serial" or "experienced entrepreneur" (N = 507). The results were qualitatively similar.

## Study 2

A limitation of the previous study is that we cannot establish whether entrepreneurship leads to the expression of positive emotions, or whether individuals who are more likely to express positive emotions self-select into entrepreneurship, giving rise to possible endogeneity. To further increase the validity of our results, we next examined how a job change from entrepreneur to non-entrepreneur and vice versa affects the emotions of an individual. To do so, we used the job history of entrepreneurs and non-entrepreneurs from Crunchbase. Crunchbase [108] is a leading online database collecting extensive information on the start-up ecosystem. Crunchbase is maintained by TechCrunch and works with a plethora of partners (venture capital firms and AngelList) to ensure that its data is accurately represented. Particularly, Crunchbase has more than 50,000 contributors and each submission of a member is reviewed by a moderator before being accepted into the database. Data from Crunchbase has been used in recent research studies [109, 110].

We combined data from Crunchbase with the tweets published by each user in order to examine the sentiment of their tweets. Based on the job history and tweets of each user, we ran a within-user (fixed-effects) analysis in order to examine how each job (entrepreneur vs. non-entrepreneur) influenced her sentiment. We excluded tweets published in 2011 or earlier for two reasons: First, Twitter became a popular communication platform after 2011 [111–113] and second, the behaviour of users is different at the beginning of a new social media platform [114].

We calculated the average sentiment of the tweets associated with each job using VADER [84]. A job change can be a transition from employee to entrepreneur or vice versa. Our sample consisted of 37,225 job profiles and 21,491,962 tweets. Our interest is in analyzing whether engagement in entrepreneurship is likely to be associated with more positive emotions.

### Dependent and independent variables

As the dependent variable we used the "average sentiment" of the tweets per job (with the sentiment of each tweet ranging from -1 (extreme negativity) to 1 (extreme positivity)). As the independent variable we used "entrepreneurship" which indicates whether a user was an entrepreneur in that job (coded as "1" for entrepreneur and "0" otherwise). We examined the job title associated with each job and identified entrepreneurs as users who had in their job

title any of the following terms: "entrepreneur", "founder", "co-founder", "business-owner", "business owner".

## Control variables

Furthermore, we controlled for several factors that could bias our results. We included several control variables from Twitter that were also used in the previous study, e.g. total number of tweets, hashtags, retweets, android source, geotagged tweets, in order to control for popularity and reputation that may affect the sentiment of each user. We also controlled for industry using 10 dummy variables—Software, Internet services, Financial services, Data and analytics, Commerce and shopping, Advertising, Health care, Hardware, Media and entertainment, Sales and marketing, with other being the excluded category.

## Analysis and results

In order to analyze the impact of job changes in and out of entrepreneurship over time we used fixed-effects regressions. This enables us to examine whether engagement in entrepreneurship is likely to be associated with the expression of more positive emotions. Specifically, based on the job history and tweets' sentiment of each user, we ran a within-user (fixed-effects) analysis using the unique identifier for each user as the grouping variable. The panel data are unbalanced as each individual does not have the same number of job changes as others. Table 11 presents the fixed-effects regression estimates. The coefficient for

**Table 11. Fixed-effects regression for sentiment.**

| Variable | Fixed-Effects Model–Sentiment |
|---|---|
| Entrepreneurship | .010 (.004) ** |
| Total_number_of_tweets (000s) | -.008 (.002) *** |
| Hashtags | .002 (.003) |
| Retweets | .014 (.006) * |
| Android_source | -.017 (.009) |
| Geotagged_tweets | -.027 (.016) |
| Software | .005 (.002) * |
| Internet_services | .001 (.002) |
| Financial_services | -.001 (.003) |
| Data_and_analytics | -.002 (.003) |
| Commerce_and_shopping | .005 (.003) |
| Advertising | -.003 (.005) |
| Health_care | -.001 (.005) |
| Hardware | -.002 (.003) |
| Media_and_entertainment | -.003 (.003) |
| Sales_and_marketing | -.001 (.004) |
| Constant | .227 (.003) *** |
| Number of observations | 37225 |
| Number of groups (users) | 25213 |
| F | 2.659*** |

Notes. The dependent variable is the average sentiment based on the Twitter user's tweets during each job. Standard errors are in parentheses.

*p < .05

** p < .01

*** p < .001.

entrepreneurship is positive and significant ($p < 0.01$), indicating that engagement in entrepreneurship is associated with more positive emotions expressed on social media.

## Discussion

Our paper examined whether entrepreneurs are more likely than non-entrepreneurs to express positive emotions on social media platforms. We also examined whether social entrepreneurs are more likely than other entrepreneurs to express positive emotions and whether serial entrepreneurs are less likely to express positive emotions on social media platforms. Using a two-study design with four samples we found that entrepreneurs express more positive emotions relative to non-entrepreneurs on social media platforms. We further showed that social entrepreneurs express more positive emotions relative to other entrepreneurs. We also find that a job change from entrepreneur to non-entrepreneur and vice versa affects the emotions of an individual on social media platforms.

Our paper makes the following contributions. Firstly, our work has implications for the limited literature looking at how social and serial entrepreneurship affects the entrepreneur. We show that social entrepreneurship can lead to more positive emotions on social media platforms than other forms of entrepreneurship, extending prior research arguing that emotion can influence social entrepreneurial intention and participation [35, 36, 39]. Further, we have argued that serial entrepreneurship can lead to less positive emotions on social media platforms although this was not significant. We proposed that a serial entrepreneur's emotions will be influenced by adverse memories of entrepreneurial failures, and fewer pleasures linked to novel experiences.

We did not previously theorize why social entrepreneurs would exhibit more positive emotions than other entrepreneurs. With the added confidence derived from observing this relation, we can tentatively propose some plausible explanations of how it operates. One explanation could be that social entrepreneurs experience altruistic enjoyment from improvements in other people's situations [115], leading to feelings of pleasure [67] which are expressed in their social media. Traditional entrepreneurs may also experience these emotions, but they are perhaps weaker than for social entrepreneurs. Another explanation could be that social entrepreneurs demonstrate that they are meeting their social duty or fulfilling norms of behaviour [116, 117], and the demonstration can bring increased public esteem, which is weaker for traditional entrepreneurs, and which is reflected in their social media. A further explanation could be that emotions are more important to the decision-making of potential investors in social enterprises, so social entrepreneurs try harder to influence the investors through their own displays of positive emotions. As these explanations are abductively derived, there is no certainty that they are accurate descriptions of the relation's operation, and further work could help to determine their veracity.

For our second contribution, we show that transition from entrepreneurship to non-entrepreneurship and vice versa affects the emotions expressed on social media platforms, supporting earlier empirical findings by Baron et al. [16] and Tata et al. [17]. As entrepreneurship generates positive emotions, the psychological compensation that entrepreneurs receive through them [50] helps to explain why entrepreneurs are willing to accept lower income and higher risks compared with people in other occupations [118, 119].

For our third contribution, we compare differences in emotions on social media platforms between entrepreneurs and non-entrepreneurs in relation to different topics. There is currently limited work examining the effect of entrepreneurship on emotions that are unrelated to the entrepreneurial process. We find that entrepreneurship positively influences emotional expressions on topics unrelated to entrepreneurship. This is important because it shows that

entrepreneurship transcends many aspects of an individual and can significantly affect an individual's well-being and mental health [24, 120].

Our study adds to recent work examining the digital footprints of entrepreneurs. For example, research has studied whether and how entrepreneurs' digital identities change in response to entrepreneurial failure by examining 760 entrepreneurs who experienced failure and their tweets before, during, and after business failure [26]. The results indicate that the financial, social, and psychological consequences of failure are reflected in entrepreneurs' tweets [26]. Obschonka et al. (2017) [48] compared the digital footprints of 57 superstar entrepreneurs and 49 superstar managers using text analysis based on the individuals' Twitter messages (called tweets) and found significant differences between the personalities of superstar entrepreneurs and managers.

Our work also allows us to draw a number of practical implications. One is that entrepreneurship can improve a person's emotional state, and their emotions directed towards people and events in general. A similar implication is drawn by Kato and Wiklund [121]. Such positive emotions can bring various favourable health outcomes [122], although the benefits of entrepreneurship are tempered by the health and personal consequences of entrepreneurial failure [72, 123]. The emotional benefits are weaker for serial entrepreneurs, in part because of their repeated exposure to possible or actual failure. However, the emotional advantages of social entrepreneurship are particularly strong, pointing to both social and personal gains from participation. Overall, there may be additional reasons for governments to encourage engagement in entrepreneurial activities beyond generating employment and economic growth.

Although positive emotions can bring benefits [122], they can also be disadvantageous in a number of ways [124, 125]. For example, positive emotions can trigger impulsivity or excessive risk-taking [126]. As entrepreneurs experience unusually positive emotions about other subjects, their emotions may unfavourably impact other aspects of their life, including health, finances, and relationships. We advise entrepreneurs to be aware of this possibility, and adapt a circumspect approach to their own evaluation of matters away from entrepreneurship.

Further, an entrepreneur's positive emotions in social media may be viewed by their readers and viewers as evidence that the entrepreneur's organisation is flourishing [13]. They may then invest in or support the organisation, regardless of its economic and operational fundamentals, potentially leading to inefficient allocation of resources as well as transfer of wealth from those investors to the entrepreneur or better-informed economic agents. Investors acting on incomplete or misperceived information has always been an issue, but with fast and direct communication by entrepreneurs through social media, and the resulting sidelining of traditional information intermediaries, the problem is accentuated. It's not an easy problem to solve; governments and social media companies struggle to control even egregious examples of misinformation, and it seems to be much harder to make people aware of their possibly incorrect inferences based on their possibly unconscious perceptions of an entrepreneur's emotional state.

There are a number of directions that may be followed in future research. Firstly, researchers could examine emotions directed towards topics divided more finely than in this paper. A "heat map" of entrepreneurial emotions over different subtopics could be built. Secondly, personal circumstances could be examined as a potential moderator between entrepreneurship and emotions. They may change the relationship between entrepreneurial outcomes and the expression of emotions. Thirdly, company performance and working environment could be examined as other potential moderators, as they may alter an entrepreneur's perceptions of likely business outcomes, and so change their emotions towards business topics. Similarly, the time that an individual has been an entrepreneur could be examined as a potential moderator,

as this feature may affect the emotions of an entrepreneur [2]. Fourthly, researchers could examine syntactic habits (e.g. number of verbs or nouns used in a sentence) of entrepreneurs through POS tagging and machine learning techniques. Finally autonomy and engagement with tasks could be examined as potential explanations of the more positive emotions expressed by entrepreneurs.

Our study suffers from many limitations. First, our results cannot be generalised to other countries. It is important to replicate the findings in developing countries and in countries where cultural norms may differ from the countries studied in the paper. Second, our study does not aim to disentangle displayed from actual emotions following previous research in this area [17]. Researchers could try to minimise the effect of entrepreneurs' communication and impression management techniques on their measured emotions, which may make entrepreneurs' reported emotions more positive than their underlying emotions [97, 127, 128]. As emotions are an internal experience, self-reports are integral to their measurement and it is a challenge to separate them from communication and impression management effects. We concur with the recommendations of Cardon et al. [45] who argue that "future research can explore how such deliberate management of the entrepreneurs' emotions impacts the outcomes with and for these stakeholders". Impression management would suggest that serial entrepreneurs, who are more experienced, would be better at managing their emotions and thus more likely to exhibit positive emotions. However, our findings are in the opposite direction, making this limitation less likely to be an issue in our study. Another limitation of our study is that "sarcasm" is challenging to be detected in online social media [129]. Finally, as two of our samples were in the entrepreneurial ecosystems of London and Los Angeles, our results cannot be generalized outside these ecosystems [130]. Furthermore, the emotional expression of users in social media platforms may be influenced by social desirability [131].

## Conclusion

To sum up, we examined differences in the emotional expression on social media platforms of traditional, social, and serial entrepreneurs, and compared them using much larger samples than have been used in previous studies of entrepreneurial emotion. We found that entrepreneurship can lead the entrepreneur to express more positive emotions relative to non-entrepreneurs. Social entrepreneurship can lead to even more positive emotions than other forms of entrepreneurship. Given the evidence here on how entrepreneurship affects emotion, and its potential to thereby impact the entrepreneur's business and other aspects of their life, we believe that further research on the influence of entrepreneurial activities on emotion will be theoretically and practically valuable.

## Author Contributions

**Conceptualization:** Nicos Nicolaou, Dimosthenis Stefanidis, George Pallis, Marios Dikaiakos.

**Data curation:** Dimosthenis Stefanidis, Hariton Efstathiades.

**Formal analysis:** Dimosthenis Stefanidis.

**Investigation:** Dimosthenis Stefanidis, George Pallis, Marios Dikaiakos.

**Methodology:** Nicos Nicolaou, Dimosthenis Stefanidis, Hariton Efstathiades, George Pallis, Marios Dikaiakos.

**Software:** Dimosthenis Stefanidis, Hariton Efstathiades.

**Supervision:** Nicos Nicolaou.

**Validation:** Dimosthenis Stefanidis, George Pallis, Marios Dikaiakos.

**Visualization:** Dimosthenis Stefanidis, Hariton Efstathiades.

**Writing – original draft:** James Waters, Nicos Nicolaou, Dimosthenis Stefanidis, Hariton Efstathiades.

**Writing – review & editing:** James Waters, Nicos Nicolaou, Dimosthenis Stefanidis, George Pallis.

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
