## [Decision Letter · Decision Letter 0]

19 Aug 2020

PONE-D-20-15982

Exploring the Sentiment of Entrepreneurs on Twitter

PLOS ONE

Dear Dr. Pallis,

Thank you for submitting your manuscript to PLOS ONE. We have received three excellent reports on your paper and I am very grateful to the highly constructive feedback the referees provided on your paper. As you will see, all three referees see merit in your study but also raise substantial concerns about your analyses, how you interpret your results, and how they fit into the existing literature. All concerns raised by the referees would need to be adressed in a major revision and a point-by-point response before we can consider publication of your work in PLOS ONE. In particular, both R1 and R2 point out that your data and analyses do not allow you to make any claims about how people actually feel or why, partly because of measurement validity issues and partly because of your study design. Furthermore, in response to R1's comments, I'd like to point out that while novelty or a theoretical contribution are not requirements for publication in PLOS ONE, we do require clearly stated research questions and appropriate discussions of your empirical results and how they fit into the existing literature.

We look forward to receiving your revised manuscript.

Kind regards,

Philipp D. Koellinger, Ph.D.

Academic Editor

PLOS ONE

2.We note that you have indicated that data from this study are available upon request. PLOS only allows data to be available upon request if there are legal or ethical restrictions on sharing data publicly. For more information on unacceptable data access restrictions, please see http://journals.plos.org/plosone/s/data-availability#loc-unacceptable-data-access-restrictions.

3. We note you have included a table to which you do not refer in the text of your manuscript. Please ensure that you refer to Table 10 in your text; if accepted, production will need this reference to link the reader to the Table.

**Comments to the Author**

1. Is the manuscript technically sound, and do the data support the conclusions?

Reviewer #1: Partly

Reviewer #2: Partly

Reviewer #3: Yes

2. Has the statistical analysis been performed appropriately and rigorously? 

Reviewer #1: Yes

Reviewer #2: Yes

Reviewer #3: Yes

3. Have the authors made all data underlying the findings in their manuscript fully available?

Reviewer #1: No

Reviewer #2: No

Reviewer #3: No

4. Is the manuscript presented in an intelligible fashion and written in standard English?

Reviewer #1: Yes

Reviewer #2: Yes

Reviewer #3: Yes

5. Review Comments to the Author

Reviewer #1: This is a well written paper that explores entrepreneurs’ emotions based on an innovative big data set from Twitter. I very much appreciate the authors’ efforts to analyze Twitter data because Twitter data can be a very valuable data source. I also like the approach of matching the Twitter data with data from Crunchbase. Nevertheless, I have some concerns about the paper in its current form:

1. My first major concern is that I don’t understand what it is you bring new to the table. In more formal words, what is (are) the gap(s) that you address with your study? At the beginning of the introduction, you motivate well the phenomenon under study and provide a detailed and extensive overview of the current knowledge of the topic. However, I am missing clear statements about what we still don’t know about entrepreneurs’ emotions, why it is important to study these gaps, and how exactly you address these gaps with your study. For me, it is absolutely essential that you state clear answers to these three points before you explain the details of your study. Therefore, I do not see any contributions of your study to the field at the moment. This assessment is further corroborated when reading your research questions. In your review of the current literature in the introduction, you can already answer those questions. Why do you need to study these questions again? To answer this question, I suggest you think mainly about theoretical reasons. It appears that you argue that your contribution is to answer these questions with an innovative data set, but this contribution would also need to be supported with a theoretical gap.

2. I am not sure I agree with some of your main statements in the paper. First, I am not sure that you actually study entrepreneurs’ emotions, but that you rather study their emotional expression on a social media platform. These expressions are likely to be influence by social desirability and you somehow need to account for this or at least be more careful and precise with your statements (see for example the book by Seth Stephens-Davidowski, 2017). In addition, expressing positive emotions on Twitter can also be a form of cognitive dissonance, i.e., entrepreneurs express positive emotions to justify their choice to become (social, serial) entrepreneurs. Second, you overstate your findings in (implicitly) drawing generalizations about all entrepreneurs when in fact you analyze entrepreneurs from London and Los Angeles. You actually analyze data from entrepreneurs in the biggest entrepreneurship ecosystems, hence you can only draw generalizations about such entrepreneurs (please see Yarkoni, 2020, about the generalizability crisis). Referring to my previous point, it may also be likely that especially these entrepreneurs express positive emotions because they are supposed to express positive emotions as entrepreneurs in one of these start-up hubs.

3. It appears that you use emotions, positive emotions, and sentiment as substitutes. While sentiment is in the paper’s title, the literature review is about positive emotions, and the measure is about emotion and sentiment as substitutes. From your description of the variable “emotion”, I am not clear what exactly you measure and which levels your dependent variable has. Please refine the description of your dependent variable and align your theoretical arguments with the operationalization. Please also note that current emotion literature argues that positive and negative emotions are no opposites, but that they are orthogonal. People can experience positive and negative emotions at the same time.

4. I wonder why you analyze London and Los Angeles separately. If they are – as you argue – similar clusters, why not analyze them together?

5. The structure of the discussion can be improved in that the contributions are not only repeated in a longer format from the introduction, but that you explain specifically how your results close gap(s) in current knowledge and advance the field.

I wish you all the best with your research and I hope my comments have been helpful.

Reviewer #2: The authors present two studies in which they investigate how types of entrepreneurs differ in their emotional expressions on Twitter. In the first study Twitter data is analyzed using VADER and uClassify. The second study combines the Twitter data with background information on the entrepreneurs to be able to investigate the endogeneity (positive people could be more likely to become entrepreneurs). I like how they make use of preprogrammed NLP functions with careful selection of meaningful data samples, count characteristics of the data and how this is integrated in their regression analysis. However, I do have some points that I would like to be clarified and improved.

1. People active on twitter are not representative of wider population. How many unsuccessful entrepreneurs are active on twitter? I can imagine that there is a strong self-selection of successful entrepreneurs. People are unlikely to be negative about a choice they have made (such as failing an entrepreneurship). Also, it seems that non-entrepreneurs are less active on twitter than entrepreneurs, which might reflect how twitter is used to promote their own business. This might drive the results.

2. You compare social with serial entrepreneurs, but also discuss traditional entrepreneurs. Why did you not include them in your analysis? Is it possible that these are the seasoned, successful entrepreneurs, as they are longer holding up the same business? This might make them score higher than the others in terms of a) success and b) pride. Both of them result in positive emotions. I also think the classification of social and serial entrepreneurs could be more sophisticated, namely using a dictionary that reflects the meaning of ‘social’ and ‘serial’.

3. Autonomy and engagement with the tasks are offered as an explanation of the effect (mid page 5), but did you try to find this in the tweets? I suspect that a relatively simple dictionary approach could give some important insights of why you see that some entrepreneurs express more positive emotions than others.

4. As all entrepreneurs are likely to experience autonomy and engagement, how is the warm glow different for social entrepreneurs? I can imagine that the warm glow also appears in entrepreneurs that feel engaged with less social causes. The way how it is posed now is subjective; social entrepreneurs can fulfill social norms, but other entrepreneurs cannot as they do not pursue social causes. However, some social groups appreciate money, leading to social norms of earning a lot of money as opposed to having a social goal in mind. To avoid this subjectivity, I think I would replace ‘warm glow’ with gratitude which you receive when you do something that others did not ask for, and that is appreciated. This is probably more often the case with social entrepreneurs than other entrepreneurs.

5. VADER: although it is successful, it is unable to handle negations, sarcasm, misspellings and certain jargon. How do you deal with this? Did you try POS tagging to dig deeper into the data?

6. I haven’t heard about uClassify before, and it sounds great! What happens if uClassify cannot find a good classification? Naïve Bayes is known for wrongly classifying those borderline cases, which are likely to happen with 10 classes. This would violate the independence assumption of Naïve Bayes. You only kept the tweets with probability higher than .9 to circumvent this problem. How many tweets were disregarded because of this rule? When I look at table 1 and 4, I see for the global sample this amounts to about 20%. With so many disregarded tweets, I can’t help but wonder how this might affect the results.

7. How many tweets per person are included? The regression in table 5 is performed at the person level, while the tables and figures before are at the tweet level. Maybe this could be accentuated when describing the data or results. Also, often some people are very active on twitter, leading to many tweets, which may bias your results. Wouldn’t it be fair to only include one tweet per person? Now you basically have multi-level data, which you treat as if it has only one level.

8. Although I appreciate study 2, in combining Twitter data with external data on entrepreneurs, I wonder why the analysis was not conducted at personal level. Now you explore the average sentiment of people on Twitter that claim to be working in a certain job. Even more, your argument resonates with the time dimension in the data, i.e. if people made shifts in their career. You claim that you investigate how these shifts are related to sentiment, but I don’t see how you integrated information on the person’s career path in the analysis. If possible, it would be nice to include a survival analysis to explore the influence of career shift on sentiment at the person level.

Small comments

1. Top page 5, second paragraph, first sentence. I think the sentence is easier to read when adjusted as such: “To begin, we argue that the act of choosing her work aims and conditions can be enjoyable in itself for an entrepreneur.”

2. Bottom page 8: “A serial entrepreneur may reflect on these unpleasant prior experiences [2], which are less likely to be considered by less experienced entrepreneurs and which impact negatively the serial entrepreneur’s emotions.” Can you add time that someone has been an entrepreneur to the analysis as control variable?

3. Table 5. Why not include the two dummy variables for serial and social entrepreneurs in one regression? Now you run two regressions to compare three groups, but you can compare both groups to the reference category (traditional) and suffice with one regression.

Reviewer #3: Dear authors,

your study presents a state-of-the art empirical analysis on the sentiment of entrepreneurs in Twitter messages. I have no issues wih the statisticak analysis and the results make sense to me. The main novelty is distinguishing between social, serial and for-profit-entrepreneurs as well as the analysis of how job changes by entrepreneurs impact emotions on Twitter.

My main problem is the treatment of prior literature. You cite a lot of general literature about the emotions of entrepreneurs but you largely ígnore the more specific (and also well-published) literature on the Twitter sentiments of entrepreneurs.

Here are a couple of papers that are very similar to your paper analyzing emotions of entrepreneurs using Twitter data and comparing them to non-entrepreneurs as well as distinguishing between failed and non-failed entrepreneurs. Please make your positioning and your treatment of prior literature more accurate when you are invited to conduct a revision.

Obschonka, M., Fisch, C., & Boyd, R. (2017). Using digital footprints in entrepreneurship research: A Twitter-based personality analysis of superstar entrepreneurs and managers. Journal of Business Venturing Insights, 8, 13-23.

Obschonka, M., & Fisch, C. (2018). Entrepreneurial personalities in political leadership. Small Business Economics, 50(4), 851-869.

Fisch, C., & Block, J. H. (2020). How does entrepreneurial failure change an entrepreneur's digital identity? Evidence from Twitter data. Journal of Business Venturing,

all the best

6. PLOS authors have the option to publish the peer review history of their article (what does this mean?). If published, this will include your full peer review and any attached files.

Reviewer #1: **Yes: **Theresa Treffers

Reviewer #2: **Yes: **Meike Morren

Reviewer #3: No

---

## [Author Response · Author response to Decision Letter 0]

5 Mar 2021

We provide a point-by-point response to the comments of the reviewers and editor in the response letter.

---

## [Decision Letter · Decision Letter 1]

13 Apr 2021

PONE-D-20-15982R1

Exploring the Sentiment of Entrepreneurs on Twitter

PLOS ONE

Dear Dr. Pallis,

Thank you for submitting your manuscript to PLOS ONE. After careful consideration, we feel that it has merit but does not fully meet PLOS ONE’s publication criteria yet. Therefore, we invite you to submit a revised version of the manuscript that addresses the points raised during the review process.

Only a minor revision is required at this point. 

We look forward to receiving your revised manuscript.

Kind regards,

Philipp D. Koellinger, Ph.D.

Academic Editor

PLOS ONE

Journal Requirements:

Reviewers' comments:

Reviewer's Responses to Questions

**Comments to the Author**

1. If the authors have adequately addressed your comments raised in a previous round of review and you feel that this manuscript is now acceptable for publication, you may indicate that here to bypass the “Comments to the Author” section, enter your conflict of interest statement in the “Confidential to Editor” section, and submit your "Accept" recommendation.

Reviewer #1: All comments have been addressed

Reviewer #2: (No Response)

Reviewer #3: All comments have been addressed

2. Is the manuscript technically sound, and do the data support the conclusions?

Reviewer #1: Yes

Reviewer #2: Yes

Reviewer #3: Yes

3. Has the statistical analysis been performed appropriately and rigorously? 

Reviewer #1: Yes

Reviewer #2: Yes

Reviewer #3: Yes

4. Have the authors made all data underlying the findings in their manuscript fully available?

Reviewer #1: No

Reviewer #2: No

Reviewer #3: Yes

5. Is the manuscript presented in an intelligible fashion and written in standard English?

Reviewer #1: Yes

Reviewer #2: Yes

Reviewer #3: Yes

6. Review Comments to the Author

Reviewer #1: Thank you very much for revising the paper! It reads very well and many constructs have become clearer. I still a few comments to consider:

1. From your introduction, it’s still not clear to me why we need your study. You state your contributions, but why are they important? For example,why is it important to examine emotion expression instead of experienced emotions? Please elaborate further on your contributions and how and why they are relevant for the field.

2. I find your RQs 2 and 3 a bit unspecific and would suggest that you split them into comparisons between the different groups of entrepreneurs, e.g. social entrepreneurs vs serial entrepreneurs. Also, are there serial social entrepreneurs? Or serial entrepreneurs that have started social AND profit-oriented ventures?

3. I’d love to see a (few) paragraph about the practical implications of our studies. What can you recommend to entrepreneurs, social media plattforms, policy makers?

I really like how you developed your paper further and wish you much success with it!

Reviewer #2: General remark: Although in many parts of the paper, the authors have toned down their claim to measure emotions, here and there this is still mentioned as such. Please read carefully the paper, and replace emotions by expressed emotions on platforms.

2. The part where you explain why social entrepreneurs express more positive emotions on social media (pages 8-10) is important, but instead of supporting your hypotheses, I would rather use this argumentation as part of the discussion where you suggest explanations (i.e. gratitude among social entrepreneurs) for the effect that you found. In the introduction you can only mention that you expect social entrepreneurs to express more positive emotions. You can mention compassion and motivations but more detailed explanations of this effect (such as gratitude) probably belong better in the discussion. The text I am referring to is:

As both traditional entrepreneurs and social entrepreneurs have substantial freedom to set their own goals and working conditions, the arguments that we presented on why traditional entrepreneurs show more positive emotions continue to apply to social entrepreneurs. However, there are additional reasons why social entrepreneurs’ emotions may be even more positive than those of traditional entrepreneurs. We propose that social entrepreneurship may lead to altruistic enjoyment of other people’s improved circumstances, and so raises the social entrepreneur’s emotions. We then argue that social entrepreneurs may receive gratitude from others as they are willing to take on the risk and effort to create positive changes in society through their initiatives, and we see how social entrepreneurship may raise social esteem and enable people to meet social norms, leading to more positive emotion.

4. The difference in expressing emotions between social and the other entrepreneurs seems to resonate with common sense (i.e. doing good rewards in good feelings) but I still struggle with the other entrepreneurs. Why do they score lower on expressing these emotions? Can you explain that? For instance, the following arguments could also apply to non-social entrepreneurs:

A social entrepreneur works to create social value in some way and can be considered to forego personal income or leisure in order to devote their time to the activity. […] Many people will try to demonstrate that they are meeting their social duty or fulfilling norms of behaviour [67, 68], and the demonstration can bring increased public esteem or protect the social entrepreneur from sanction for neglecting their societal duties.

I can imagine that the other entrepreneurs feel the same way about showing off profit or other entrepreneurial success. All in all, I agree with your argumentation, but I would like to challenge you to come up with an explanation for why the other entrepreneurs express fewer positive emotions. As R1 suggests, this might also be due to social desirability. If social entrepreneurs score higher on social desirability, then this could also be an alternative explanation. I think a more critical note to the explanation of the effect that is found should be mentioned in the discussion.

6. I appreciate the additional analysis; it is good to know that your results are robust. I do observe that the t values are much higher for all tweets, than only those who have a probability by uClassify of .9 or higher. Could this mean that uClassify has issues with classifying tweets that are very strong in expressing emotions between entrepreneurs and non-entrepreneurs? Why is that? Could you add a remark in the conclusions?

As for the other comments, I feel like you addressed them satisfactorily, thank you for your efforts to improve the paper!

Reviewer #3: The authors have done a good job. Well done. I am happy to recommend this revised version to be published.

7. PLOS authors have the option to publish the peer review history of their article (what does this mean?). If published, this will include your full peer review and any attached files.

---

## [Author Response · Author response to Decision Letter 1]

10 Jun 2021

Please see the response letter document.

---

## [Decision Letter · Decision Letter 2]

28 Jun 2021

Exploring the Sentiment of Entrepreneurs on Twitter

PONE-D-20-15982R2

Dear Dr. Pallis,

We’re pleased to inform you that your manuscript has been judged scientifically suitable for publication and will be formally accepted for publication once it meets all outstanding technical requirements.

Kind regards,

Philipp D. Koellinger, Ph.D.

Academic Editor

PLOS ONE

Additional Editor Comments (optional):

Reviewers' comments:

Reviewer's Responses to Questions

**Comments to the Author**

1. If the authors have adequately addressed your comments raised in a previous round of review and you feel that this manuscript is now acceptable for publication, you may indicate that here to bypass the “Comments to the Author” section, enter your conflict of interest statement in the “Confidential to Editor” section, and submit your "Accept" recommendation.

Reviewer #2: All comments have been addressed

2. Is the manuscript technically sound, and do the data support the conclusions?

Reviewer #2: Yes

3. Has the statistical analysis been performed appropriately and rigorously? 

Reviewer #2: Yes

4. Have the authors made all data underlying the findings in their manuscript fully available?

Reviewer #2: No

5. Is the manuscript presented in an intelligible fashion and written in standard English?

Reviewer #2: Yes

6. Review Comments to the Author

Reviewer #2: Thank you for your efforts throughout the review process. As a sidenote, it might be useful to consider to present the data and (part of) the code you've used for other researchers on a platform such as GitHub.

7. PLOS authors have the option to publish the peer review history of their article (what does this mean?). If published, this will include your full peer review and any attached files.

Reviewer #2: **Yes: **Meike Morren

---

## [Editor Report · Acceptance letter]

22 Jul 2021

PONE-D-20-15982R2 

Exploring the Sentiment of Entrepreneurs on Twitter 

Dear Dr. Pallis:

I'm pleased to inform you that your manuscript has been deemed suitable for publication in PLOS ONE. Congratulations! Your manuscript is now with our production department. 

Kind regards, 

on behalf of

Dr. Philipp D. Koellinger 

Academic Editor

PLOS ONE